# Advances in Genomics-Based Breeding of Barley: Molecular Tools and Genomic Databases

Asad Riaz [1,†], Farah Kanwal [1,†], Andreas Börner [2,3], Klaus Pillen [3], Fei Dai [1,*] and Ahmad M. Alqudah [3,*]

1   College of Agriculture and Biotechnology, Zhejiang University, Hangzhou 310058, China; asad.riaz76@gmail.com (A.R.); 11816105@zju.edu.cn (F.K.)
2   Leibniz Institute of Plant Genetics and Crop Plant Research (IPK), Corrensstraße 3, 06466 Seeland, Gatersleben, Germany; boerner@ipk-gatersleben.de
3   Institute of Agricultural and Nutritional Sciences, Martin Luther University Halle-Wittenberg, Betty-Heimann-Straße 3, 06120 Halle (Saale), Germany; klaus.pillen@landw.uni-halle.de
*   Correspondence: daifeijx@163.com (F.D.); ahmad.alqudah@landw.uni-halle.de (A.M.A.)
†   Authors contributed equally.

**Abstract:** Barley is the fourth most important cereal crop and has been domesticated and cultivated for more than 10,000 years. Breeding climate-smart and stress-tolerant cultivars is considered the most suitable way to accelerate barley improvement. However, the conventional breeding framework needs to be changed to facilitate genomics-based breeding of barley. The continuous progress in genomics has opened up new avenues and tools that are promising for making barley breeding more precise and efficient. For instance, reference genome assemblies in combination with germplasm sequencing to delineate breeding have led to the development of more efficient barley cultivars. Genetic analysis, such as QTL mapping and GWAS studies using sequencing approaches, have led to the identification of molecular markers, genomic regions and novel genes associated with the agronomic traits of barley. Furthermore, SNP marker technologies and haplotype-based GWAS have become the most applied methods for supporting molecular breeding in barley. The genetic information is also used for high-efficiency gene editing by means of CRISPR-Cas9 technology, the best example of which is the cv. Golden Promise. In this review, we summarize the genomic databases that have been developed for barley and explain how the genetic resources of the reference genome, the available state-of-the-art bioinformatics tools, and the most recent assembly of a barley pan-genome will boost the genomics-based breeding for barley improvement.

**Keywords:** barley; genome; next-generation sequencing; databases; genomic breeding

## 1. Introduction

Barley (*Hordeum vulgare*, 2n = 2x = 14)—domesticated from its wild relative, *Hordeum spontaneum,* which was found at archaeological sites in the Fertile Crescent over 10,000 years ago—and was the first crop cultivated by humans [1], and today, it ranks as the fourth largest cereal crop in terms of planting area (http://faostat.fao.org, accessed on 21 January 2021). It is a major food source in some developing countries [2], as it can tolerate more environmental stresses than wheat and other cereals [3].

Concentrating on the important agronomic traits of barley, such as the number of tillers [4], grain number [5], plant height [6], disease resistance [7], abiotic stress tolerance [8], and malting quality [9], plant breeders have made efforts towards advanced molecular breeding in order to attain the best combination of traits for satisfying farmers' and consumers' demands.

Next-generation sequencing (NGS) technology has sped up the progress of the genome sequencing and re-sequencing of cereal crops, with huge potential for making a remarkable impact on breeding [10]. The published genome sequences of rice [11], wheat [12], maize [13], barley [14], and other cereal crops have supported researchers in determining

the genetic and physical mapping of molecular markers in specific loci/genes. These identified markers can be applied on the basis of genotyping technology to conduct molecular marker-assisted selection (MAS) breeding or to determine the genetic relationships among diverse accessions. The reference genomes are used for re-sequencing to frame a bulk segregation analysis (BSA) of individuals and to analyze sequence diversity at the genomic level [15]. The BSA-Seq combination of whole-genome sequencing (WGS) has been applied for quick identification of mutations (nucleotide changes) using MutMap and quantitative trait loci (QTL) in major cereals [16]. Some other sequencing-based genomic approaches that have been utilized in barley include bulk segregant ribonucleic acid (RNA) sequencing (BSR-seq) [17], specific-length amplified fragment sequencing (SLAF-seq) [18], and genome-wide association scan (GWAS) [19]. The development of transcriptome sequences has improved the interpretation of genes with an understanding of the domestication and regulation of gene function networks using their expression patterns. In addition to the identification of genetic markers and the availability of published genomes, clustered regularly interspaced short palindromic repeats-associated protein 9 (CRISPR/Cas9) is promising for application to modern breeding and is a novel technology for genome editing in major cereals [20]. CRISPR/Cas9-based directional breeding is highly efficient and saves more time than other breeding techniques that use genome editing [21].

Here, we review the advances in the application of NGS in barley breeding and outline the applications of genome editing in modern breeding. Moreover, this review summarizes the genomics-based approaches to gene identification and sheds light on the availability of genomic resources and databases for barley. This effort can provide a theoretical groundwork which will help to develop knowledge-based strategies for further adaptation of barley to our needs.

## 2. Brief Description of the Available Genomic/Transcriptomic Information and Databases

The open availability of sequencing data of barley has enhanced the understanding of the genetic and regulatory functions of genes related to agronomically important phenotypes. The genome size of barley is almost 5100 Mb (~5.1 Gb) [22], and the first reference genome was established by the International Barley Sequencing Consortium (IBSC) [23]. There are several databases available for barley genome sequence data (see Figure 1), which contain different information and offer different tools for the interpretation of genomic resources and analysis.

### 2.1. EnsemblPlants

EnsemblPlants is a web database that acquires the genomic and proteomic data of different plant species, including barley [24], and also offers access for application of the Basic Local Alignment Search Tool (BLAST) to check the index of similarity of the queried sequence with the barley genome. It manages data with the collaboration of the Gramene Database [25].

### 2.2. Nord-Gen

Nord-Gen (Nordic Genetic Resource Center) is an international database for genetic stock and mutant data collection that is centered in Sweden. Now, it is a gene bank as well as a center for genetic resources. It houses information on barley genes, mutants, and gene nomenclature.

### 2.3. BARLEX

BARLEX presents the first linearly ordered barley sequence and provides physical and genetic maps of molecular markers and genes using different version of assembly and gene set, with the expression profiling data of 16 developmental stages, as well as exome capture data [14].

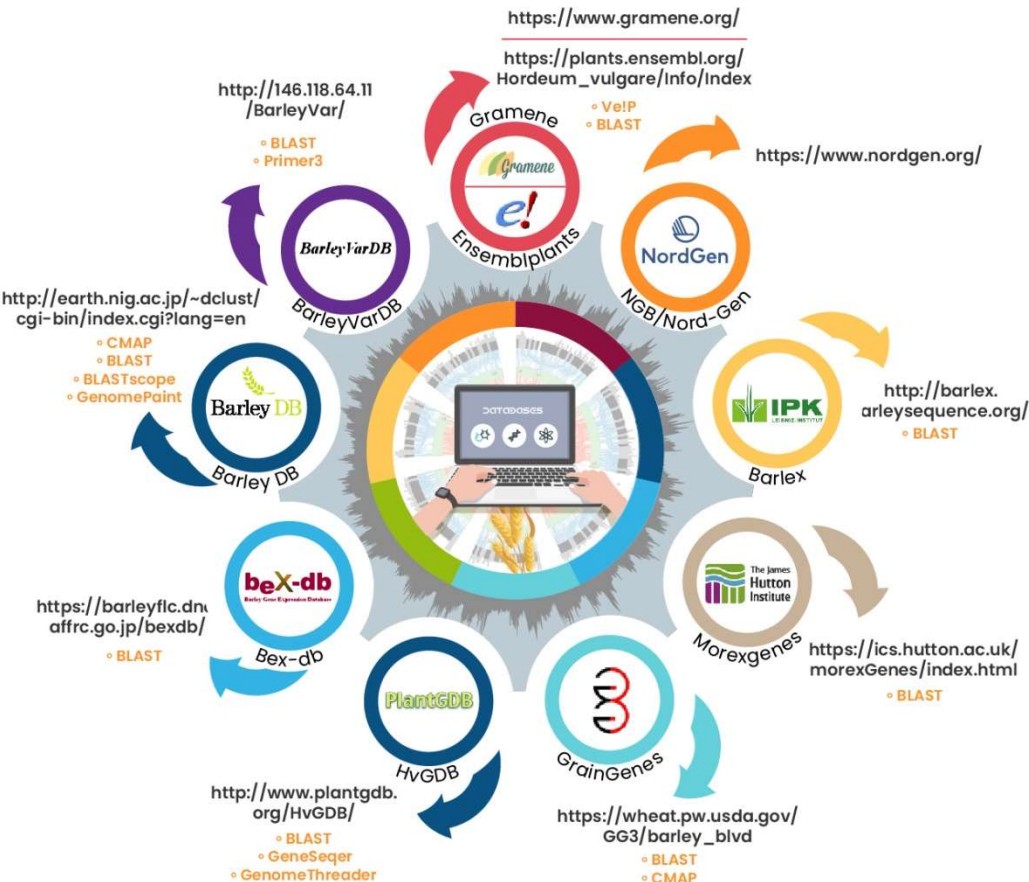

**Figure 1.** Compilation of quick and short information on the major barley genome databases. Each small, colored circle introduces a different database (name under each circle), with the arrows pointing to the addresses of their webpages. The yellow-colored text under the webpage links shows the names of the tools offered in that database.

### 2.4. MorexGenes

MorexGenes offers access to gene expression levels from the RNA-seq data of the barley cultivar, Morex, which are assembled from whole-genome shotgun sequences of Morex. It also contains a BLAST tool for carrying out basic alignments [26,27].

### 2.5. GrainGenes

GrainGenes is a genetic database primarily containing data on barley and wheat, such as genetic markers, gene expression, and QTLs. It also provides tools for BLAST, genome-specific primer design, and a genetic map display/visualizer [28].

### 2.6. HvGDB

HvGDB is a barley database provided by PlantGDB (Plant Genome DataBase) that offers a focus on comparative genomics by using genomic data integration and analysis. It contains advanced tools for comparative genomics, such as CrowsNest, which is used to analyze syntenic relationships among grass genomes.

### 2.7. Bex-DB

Bex-DB was developed by the National Institute of Agrobiological Sciences (NIAS) with the availability of full-length cDNA libraries of a two-rowed malting barley, Haruna Nijo. It offers BLAST, a genome viewer for IBSC, and comprehensive analysis of gene expression data [22].

### 2.8. BarleyDB

BarleyDB includes material on barley germplasms and genome resources, as well as BLAST and extra tools (http://shigen.nig.ac.jp/shigen/tool/tool.jsp?lang=en, accessed on 16 January 2021), such as BLASTscope, which enables the creation of graphical figures of BLAST query results, and GenomePaint, which enables the creation of graphical figures (circular or linear) of a specified genomic region.

### 2.9. BarleyVarDB

BarleyVarDB is a recently established database that provides data related to barley's genomic variations in the form of three datasets—SNPs, InDels, and whole-genome sequences of wild (eight accessions) and cultivated (13 accessions) barley genomes—with a web-based application of BLAST with Primer3 [29].

### 3. Mapping and Identification of Useful Genomic Regions/Genes Using the Established Genetic Information/Genomic Resources

With the completion of the barley genome sequence and the open availability of the databases, barley breeding is now in the "genomic" era, while barley research is in the post-genomic era. Genome-based identification and utilization have progressively become the core techniques for identifying gene functions. Several genomic databases for barley have been established, and they are being utilized in different ways to identify or map the specific genes or genomic regions. The main approaches for the genome-based association studies utilized in barley are shown in Figure 2 and discussed below.

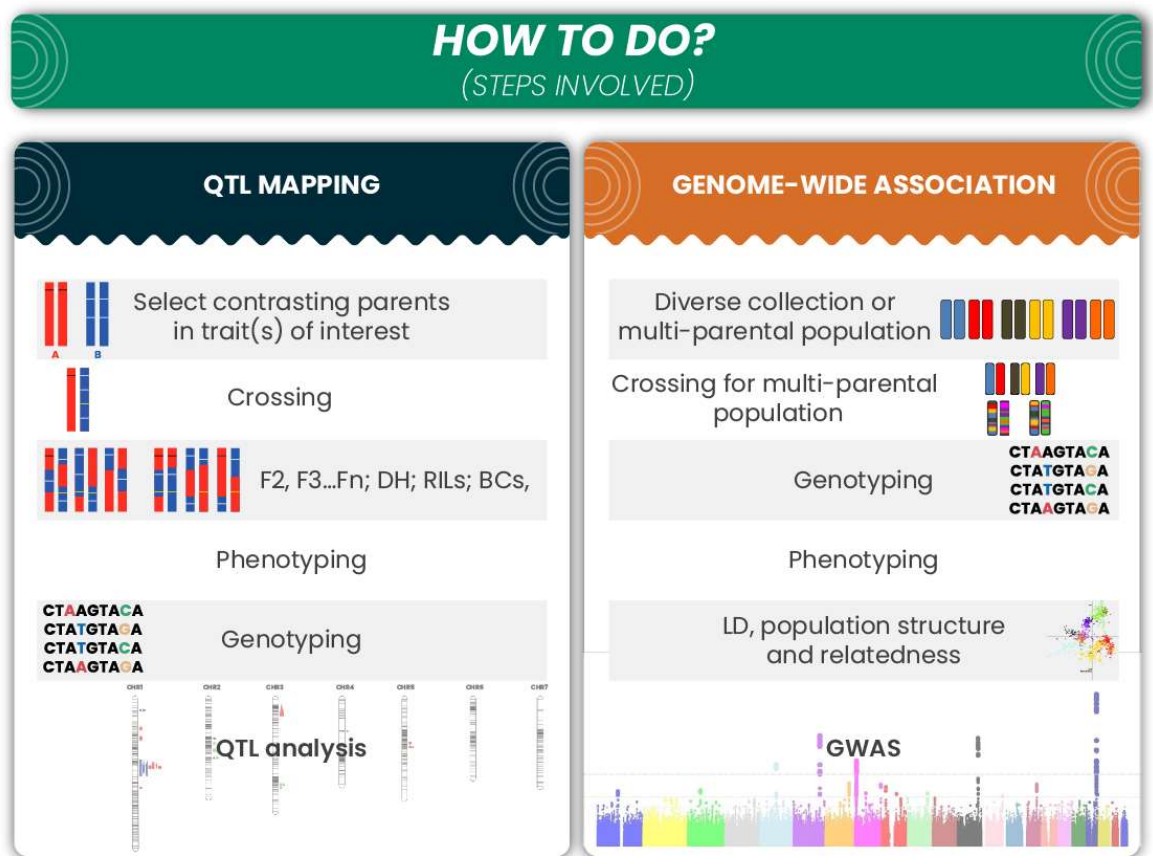

**Figure 2.** Brief description of the major steps involved in QTL mapping and GWAS.

### 3.1. Quantitative Trait Locus (QTL) Mapping

QTL mapping is a statistical-analysis-based technique that integrates the phenotype of a specific trait (phenotypic data) with molecular markers (genotypic data) in a developed

population to determine a genetic region (QTL) in complex characters [30]. A number of QTL statistical models have been developed, such as standard interval mapping (SIM) and multiple imputation (IMP), which are used when the single QTL is unlinked, and composite interval mapping (CIM), which is designed to map the genetic linkage for both linked and unlinked QTLs/genes on the chromosome. The performance of these methods is measured based on the calculated LOD (logarithms-of-odds) scores, and QTLs are usually considered significant above the threshold LOD score of 3.0. Open access to reference genomes helps researchers by providing genetic information on the genes responsible for QTLs, which are the target of MAS. Traditional QTL mapping requires a balanced population with known recombination data. Following this, a statistical association can be inferred between phenotypic and genotypic data through linkage mapping [31,32]. The location of a QTL can be determined where the allelic variants of a physically or genetically linked molecular marker display a significant effect on a quantitative trait in the studied population [32]. The identification of genomic locations facilitates the further identification of responsible genes and the exploration of the mechanism of genetic variation [33]. Like in other crop plants, the linkage or QTL mapping approach is widely applied in barley, and its power to identify QTLs that control target traits in a specific population has been proven. In barley, dozens of populations have been developed and used for QTL studies using genetic markers, either RFLPs [34,35] or SSRs [18,36]. NGS dramatically improves the density of SNPs; thus, researchers are empowered to detect QTLs for various traits. A sets of 47 introgression lines were used to map drought-tolerance-related QTLs, and 11 out 44 QTLs were found to be involved in the growth rate and water-use efficiency [37]. A population of 100 recombinant inbred lines (RILs) of barley derived from a cross between Syrian and European parents was used to identify the grain-yield-related QTLs, and a total of 60 QTLs were mapped, with the largest number on 2H associated with the heading [38]. The mapping of a population of 93 recombinant inbred lines (RILs) developed from a cross between the "Rasmusson" cultivar, which was moderately susceptible to fusarium head blight, and a highly susceptible Japanese landrace was used to identify the QTLs associated with susceptibility to disease in barley, which resulted in a total of six QTLs being identified on the 2H, 5H, 6H, and 7H chromosomes [39]. For nutrient-use efficiency, 17 QTLs associated with phosphorus (P) acquisition and P-use efficiency [40] were identified, and 15 QTLs related to nitrogen-use efficiency under low nitrogen were identified in 94 recombinant inbred lines (RILs) of the Prisma × Apex mapping population [41,42]. Similarly, a number of QTLs related to malting quality were identified [43,44], and phenolic compounds were found to be associated with agronomic traits [45]. As an example of NGS application in barley, genotyping by sequencing (GBS) was used for QTL analysis in a family of recombinant inbred lines (RILs) to detect the *Breviaristatum-e* (*ari-e*) locus [46]. QTL mapping remains a strong approach that is recommended for identifying QTLs in barley, especially with the recent advancements in NGS. QTL mapping requires genetically diverse biparental segregating populations, and the diversity of the populations may affect the detected QTLs. In general, QTL mapping shows genomic regions that affect the genotype through loci associated with a trait, but is unable to identify the specific genomic loci, i.e., SNPs. The limitations of QTL analysis can be overcome by using GWAS, which can narrow down the candidate genomic regions using naturally diverse populations based on linkage disequilibrium (LD).

### 3.2. Genome-Wide Association Study (GWAS)

Like QTL mapping, GWAS also uses statistical association mapping (AM) between the molecular marker and the trait of interest. With diverse populations and based on LD without known recombination, the historical recombination can be handled. The LD could be the result of physical linkage, as well as genetic drift, selection after mutation, and population structure [47]. GWAS interprets the associations of each marker and trait of interest, which are evaluated using the individuals of a diverse population [48]. A major problem in AM is the control of false positives, which can arise due to the population

structure and family relatedness. False positives are often controlled by incorporating covariates for structure and kinship in mixed linear models (MLMs). These MLM-based methods are single-locus models and can introduce false negatives. A number of statistical tools ((MLM, compressed MLM (CMLM), ECMLM, multi-locus mixed model (MLMM), general linear model (GLM), and fixed and random model circulating-probability unification (FarmCPU)) were applied for AM in order to find significant markers. It was reported that the FarmCPU-based GWAS model could perform better compared to the other models, as it efficiently controlled for false-positive associations [49]. With the fast growth and availability of sequencing technologies, GWAS is now a prevailing tool for determining the loci underlying the natural variations in different traits of crops [48]. In GWAS, the population needs to be genotyped once; subsequently, it can be utilized repeatedly for the mapping of different traits using new phenotypic data [48]. GWAS has the limitation of its high rates of false positives due to population structures and genetic relationships.

GWAS studies have been conducted in barley for more than a decade [50]. The recently developed 9K and 50K iSelect SNP array has strongly enhanced the efficiency of the GWAS tool for novel QTLs/gene detection in barley [51]. For instance, gene targets important agronomic traits, such as the *HvCO-like* genes; some novel QTLs associated with the heading were identified with the application of GWAS in a mixed spring barley population with photoperiod sensitivity and reduced photoperiod sensitivity [52]. A population of 218 accessions of spring barley (mixed two-row and six-row) was employed in a GWAS-based analysis for the genetic dissection of the effect of the row type on the number of productive tillers [6]. In another GWAS-based analysis, some yield-related traits were examined to identify associated novel genes in a population of 615 barley cultivars, and two novel chromosomal associations with seed germination were found [53]. In addition, GWAS analyses of disease resistance against, for example, spot blotch, leaf rust, and stripe rust [54], as well as for malting and beer quality traits, were successfully carried out [55]. In addition to natural populations, family-based populations, such as the nested association mapping (NAM) population [56] and multi-parent advanced generation intercross (MAGIC) populations [57], were analyzed through GWAS in barley. They proved their utility in uncovering the basis of key agronomic traits in barley [58,59]. The most frequently used markers for GWAS are SNPs, but the exploration of the complex relationships between quantitative phenotypes and biallelic SNPs is limited [60,61]. Later, this limitation can be overcome through the analysis of haplotype blocks in targeted regions associated with complex traits [62]. A recent study was conducted to compare single-SNP-, multi SNP-, and haplotype-based GWAS analyses in barley, and much better results were found with the construction of haplotype blocks [63]. A number of studies have focused on the application of GWAS in barley [64–66]. In a recently presented review, we described the genetic discoveries in barley and provided a layout about how the GWAS tools can be utilized in barley breeding programs [67].

### 3.3. Integration Bridge: A Way to Overcome the Limitations of QTL Mapping and GWAS

Considering the limitations of each approach, an integrative bridge between both techniques can be a powerful approach for the genetic dissection and identification of the loci associated with a trait of interest. This combination compensates for the limitations resulting from false positives and facilitates the detection of rare or small-effect QTLs with high-resolution identification [68]. The output of GWAS is considered an excellent step for selecting true segregating parents in order to develop populations depending on their contrasting situations in specific phenotypes and genomic regions (allele(s)). The detection of genomic regions for the same trait in both populations with these mapping strategies is a genetic validation of the QTLs. Hence, it is best to combine both mapping strategies for the most accurate QTL results, which can be used for further genetic and molecular analyses. A rapid detection of loci responsible for complex traits was shown by using this approach in rice; 200 rice varieties in an association population together with 192 RILs were used, and reliable loci that were responsible for seed vigor were identified with the simultaneous

application of QTL mapping and GWAS [68]. In barley, the combination of both mapping populations has not yet been applied, but it is a promising approach that constitutes a step forward in genetic analysis and in the identification of candidate genes.

*3.4. Genome-Wide Analysis or Identification of Gene Families*

The robust sequencing technology and openly available genome databases provide a great opportunity for researchers in genomic analysis. Bioinformaticians are focusing on the development of new methods for analyzing the available genomic datasets [69]. A number of strategies and tools have been developed for comparative genomics or genome-wide analysis [70–72]. Of these, genome-wide identification has been applied to identify the members of specific gene families of transcription factors by using the reference genomes and phylogenetic analysis of closely related species [73]. This provides potential insights for exploring the regulatory mechanism and functional foundation of a gene family that encodes a specific protein. Several strategies are applied in such studies, including the following three: (i) identification of gene family members using gene annotations, which requires a large genome, and the annotation should be correct, as an error in the annotation raises the chances of false-positive sequences; (ii) family members can be identified by using the BLAST tool in public databases; the query sequences are usually from model species, e.g., Arabidopsis; this could result in fewer family members being identified due to the presence of species-specific genes, but it is beneficial to identify gene family members with non-canonical domains; (iii) gene family members can be identified using the HMMER program [74], which is based on hidden Markov models; this program can be used to generate a file (HMM) of gene families, and it can also identify distant gene family members with a better gene representation [74]. Several studies have been conducted, and they focused on genome-wide analysis of the evolution, identification, and regulatory network, and expression of gene families in barley, such as the signal transduction cascade (MAPK/KK/KKK; mitogen-activated protein kinase/kinase kinases/kinase kinase kinases) involved in biotic and abiotic stresses, were subjected to genome-wide identification, and 20 MAPKs, six MAPKKs, and 156 MAPKKKs of the MAPK family were identified in barley [75]. The potential function of nuclear factor-Ys (NF-Ys), which facilitate salt-stress tolerance, was determined by identifying the co-expression of 23 members in barley [76]. The epidermal wax-related stress resistance is associated with β-ketoacyl CoA synthetase (KCS) genes, and 33 KCS gene family members were found to be evenly distributed in barley chromosomes. [77]. The Hsp20 gene family, which is associated with heat-shock tolerance, has 38 putative members in barley [78]. Similarly, genes regulated under heavy metal stress were identified using barley transcriptomic data [79], and plant-phytohormone-related gene families that are involved in different developmental stages were also subjected to these bioinformatics-based gene identification techniques [80].

## 4. Genome-Based Molecular Breeding

The progress in genomics and access to publicly available genomic databases has enhanced existing breeding methods, which has facilitated the development of novel approaches to barley breeding. Modern breeding is basically a strategy for evaluating the genetic gain of a new genotype by separating the genetic effects from the environmental and noise components [81]. Plant breeding is based on different strategies, such as traditional selection by using phenotypic data for genetic evaluation [81], MAS based on specific genetic markers associated with the relevant trait, where individuals are selected based on their marker scores [82], or genomic selection, which is an advanced way of selecting individuals based on genetic markers with small effects on phenotypic variation. The steps involved in the process of genome-based molecular breeding are shown in Figure 3. MAS breeding has numerous advantages over traditional breeding, such as the lack of the need to permanently generate phenotypic data if traits are complex, its cost effectiveness, and time consumption, which impedes phenotyping, for example, with malting quality. MAS is a quick process that does not require the phenotype testing of huge progeny sets, and

pyramiding of multiple alleles is possible [83]. In addition, the linkage drag is reduced [84], and the genetic gain is increased compared to the use of phenotypic selection [85]. The genetic merit of an individual can be evaluated by using a larger population size without compromising the genetic gain by narrowing the genetic diversity [85]. MAS-based breeding programs have been extensively conducted in other cereals like wheat and rice, and it is suggested to further utilize this breeding application in barley for yield-related and stress-tolerant traits, as MAS has proven to be successful in barley, with the identification of elite lines of improved malting quality by stable transfer of the thermostable β-amylase from wild barley into a commercial variety [86]. With the development of NGS technologies, the number of markers has been increased, and the breeding efficiency has significantly increased. SSRs, SNPs, InDels, and haplotypes have become the most important markers to use for efficient genotyping and construction of genetic maps. For example, SSR markers were employed to map the Als gene on the 3H chromosome of barley, and it was found to be responsible for the low number of tillers [87]. In a study of barley, a total of 83 significant marker–trait associations were found to be associated with six different yield-related traits under drought conditions [88]. A similar thousand-SNP-marker set was used for association mapping of salt tolerance in barley [89]. A semi-dwarfness gene in barley, ari-e, was mapped using SNPs and InDels in a 10 Mb to 0.58 Mb interval on the POPSEQ physical map [90]. Different strategies (Sections 3.1 and 3.2) have been applied for the identification of QTLs of relevant traits in barley for marker identification and utilization, which strongly enhanced the genomic-based molecular breeding of barley, e.g., through MAS. Genome-wide selection is also known as genomic selection (GS) and is a new breeding strategy with potential for significant outcomes in plant breeding [91]. In GS, a genetically diverse test population is thoroughly genotyped and phenotyped to predict its phenotypic performance based on genomically estimated breeding values (GEBVs). The large breeding population is then genotyped, and the GEBVs are used to predict the phenotypes of the lines of the population. GS has emerged as a valuable tool for improving complex traits that are controlled by QTLs with small effects. Various simulation models for predicting the selection accuracy depend largely on the marker density, marker type, size of training populations, and trait heritability. Compared with QTL mapping and GWAS, GS has more promise for harnessing genetic gains from genetic resources for quantitative traits, and it is seen as a more reliable and useful approach [92]. It has been applied in barley breeding, such as in a study conducted by the University of Minnesota, where six-row barley lines were evaluated for GS; a significant gain in grain yield of 186.1 kg/ha was obtained, and 1.85 ppm of deoxynivalenol was observed, which is associated with malting quality [93].

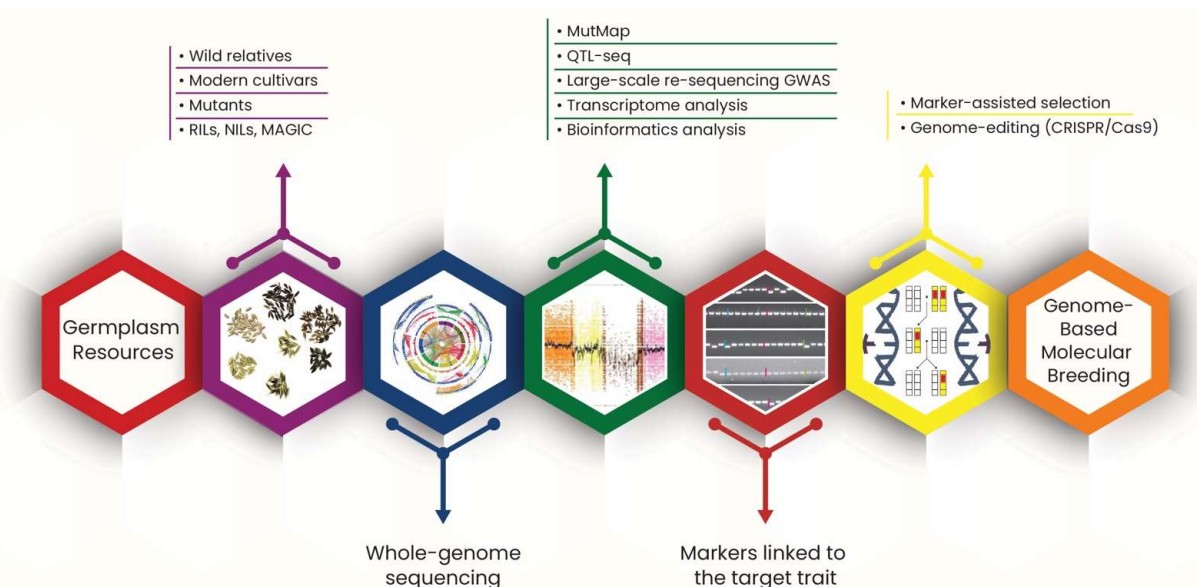

**Figure 3.** A schematic infographic explaining the process of genome-based molecular breeding.

### 5. Genome Editing, Characterization, and Functional Validation of Identified Genes

Genome editing has a history of challenges, particularly in complex genomic structures in plant molecular biology. The modern application of genome editing could support traditional breeding without developing the transgenes by overcoming the limit of mutagenesis, which induces a random mutation that is not always useful. In crops, there are several methods of performing mutagenesis in order to develop mutant material for genetic studies. Some of them are unclear about their underlying genetic mechanisms, which can be overcome by modern genome editing [94]. In genome editing, three major technologies are (i) zinc finger nuclease (ZFN), where an endonuclease linked with a multi-zinc-finger DNA-binding domain specifically recognizes and cuts target DNA [95], (ii) transcription-activator-like effector nuclease (TALEN), where multiple transcription-factor-like domains bind an endonuclease domain that recognizes target DNA sequences [96], and (iii) the widely used CRISPR/Cas9, where multiple genes can be specifically targeted by using synthetic guide RNA (sgRNA), which can be easily constructed with a chemical synthesis method [97] using chimeric sgRNA or dual RNA (crRNA:tracrRNA) [98].

CRISPR-Cas9 is, so far, the most promising and versatile genome-editing technology; the Cas protein uses the sgRNA to bind to a targeted DNA site, followed by dsDNA cutting [99]. The DNA break can be repaired in two ways: a non-homologous end-joining (NHEJ) repair that creates random insertions and deletions (indels), resulting in mutations through targeted gene knock-outs [100,101], and a homology-directed (HR) repair, which is more precise in the exchange of homologous sequences, resulting in knocked-in genes [102]. Gene-editing knowledge may be used in two ways. First, the detailed molecular and physiological study of edited knock-out and knock-in mutants can assist in further explaining the molecular functions and interactions of genes that are important in barley breeding. Second, edited genes that exhibit a proven trait-improving effect without adverse side effects may be introduced into barley breeding programs, provided that the national legal regulations permit the release of genome-edited cultivars. In barley, target genes can be knocked out after selecting the appropriate sgRNA sequence from available genome sequence databases or after re-sequencing the target gene from a particular barley cultivar [103,104]. Numerous genome-editing studies have been carried out in barley with CRISPR/Cas9, such as the increase in phytase activity in seeds by stacking the PAPHY_a gene [105], regulation of the cytokinin metabolism gene *HvCKX1/3* [106], validation of the 2OGO gene responsible for fusarium head blight disease [107], functional dissection of biosynthesis of vitamin-E-related genes (*HGGT/HPT*) [108], and validation of the viral-resistant gene *HvMORC1* in barley [109].

### 6. Conclusions

Despite steady progress in barley breeding, there is still a great need for improving barley cultivars that are adapted to diverse growing conditions. Recent progress in genomics research has provided geneticists, biologists, and breeders with a number of modern tools and technologies that impart precision and efficiency to breeding programs. The assembly of the first barley reference genome offered certain opportunities for the application of genomics in plant breeding. Several molecular, bioinformatics-, and genomics-based approaches that use genomic information in combination with sequencing, re-sequencing, and genotyping datasets are being utilized to study important agricultural traits and their linked genes. Genomic annotation of barley faces the problem of unclear functional information; for example, knowledge of the biochemical activities of many agronomic-trait-related genes is lacking, which could be inferable from proteins encoded with specific domains. A major challenge is to (i) functionally characterize the genes linked to molecular and morphological traits associated with variant forms and (ii) annotate the functional data of the causative genes in the appropriate gene databases. Notwithstanding these challenges, the continuous improvement of gene-editing technologies, with the best example being CRISPR/Cas9, has provided a strong foundation for overcoming these limitations. The recently developed high-quality genome assembly of the Golden Promise

cultivar is an emerging focus for genome-editing experiments by the barley research community. Future sequencing technologies should drive the further improvement of available reference assemblies and sequence additional barley cultivars and wild barley accessions. This will perfectly facilitate the development of a definitive catalogue of genomic diversity information with large-scale variation and the identification of a rich source of usable genes.

**Author Contributions:** A.R. and F.K. wrote the manuscript; A.R., K.P., and A.B. revised the manuscript. F.D. and A.M.A. designed and supervised the study. All authors have read and agreed to the published version of the manuscript.

**Funding:** This work was funded by the National Key R&D Program of China (2018YFD1000706), the National Natural Science Foundation of China (31871607), and the German Federal Ministry of Research and Education (BMBF) IPAS grant BARLEY-DIVERSITY (FZ 031A352A).

**Acknowledgments:** We are thankful to Iftikhar Mehdi for helping with the figure designs.

**Conflicts of Interest:** The authors have no conflict of interest to declare. We certify that the submission is original work and is not under review by any other publication.

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
