# Peer review of "Advances in Genomics-Based Breeding of Barley: Molecular Tools and Genomic Databases"

_agronomy, doi:10.3390/agronomy11050894_

Round 1

Reviewer 1 Report

The manuscript aims to describe the different technologies used to identify genes useful for barley breeding and some breeding techniques that use these technologies.

The title does not reflect the content of the manuscript.

Many approximations and inaccuracies are present in the text, with improper use of many definitions.

In each paragraph the technologies used are described very extensively without giving any reference and little space is given to their applications in barley. The latter are only mentioned at the end of each sub-section and do not even represent all the studies carried out applying the various technologies. Therefore, the text must be organized differently giving more space to the applications of the various technologies in the barley breeding and paying attention to the correctness of the statements. 

Summary sub-section is really a conclusion and it must be completely rewritten and so must the abstract, in accordance with the revision of the entire text.

References must be corrected in according to the Instructions for authors. A very extensive correction of English by native speakers is required.

Author Response

Dear Reviewer,

We are grateful to you for your time and constructive comments on our manuscript. We have implemented all comments and suggestions in the revised version of the manuscript. Changes in the initial version of the manuscript are highlighted as TRACK CHANGES in the revised version. 

We look forward to receiving a positive response from you.

Yours sincerely,

On behalf of the co-authors

Ahmad Alqudah, PhD

Reviewer 2 Report

Broad comments:

Riaz et al. summarized the genomic resources developed for barley and discussed the progress of modern genomics‐based breeding in barley with emerging genomic tools, which provide new theoretical insights into next‐generation molecular breeding for sustainable barley improvement. I think it is a well-written review article that summarizes the progress that has been made for the molecular tools at the disposal of a barley breeder.

Specific comments:

Line 13: Barley not Barely

Line 14: I think the terms should be written in a certain way. For example next generation sequencing in line 14 and Next-Generation Sequencing in line 46.

Line 35: I think it is better like “is the earliest crop which cultivated by humans”

Line 37: maybe “developing” countries instead of “poorer”

Line 164: A number of statistical tools [MLM, Compressed MLM (CMLM), ECMLM, Multi Locus Mixed Model (MLMM), General Linear Model (GLM) and Fixed and random model Circulating Probability Unification (FarmCPU)] were applied for AM to find significant markers.

Line 84 and 193: I think there should not be paragraph empty space before the word Figure.

References: after the name of the magazine a dot is inserted in all references. For example, in line 363 “Purugganan, M.D.; Fuller, D.Q. The nature of selection during plant domestication. Nature. 2009, 457, 843‐848.”

Line 386: Oryza sativa

Line 387: Triticum aestivum

432: Hordeum vulgare

439: Hordeum vulgare

454: Hordeum vulgare

459: H. vulgare

470: Hordeum vulgare

512: Oryza sativa

531: Hordeum vulgare

534: Hordeum vulgare

552: Hordeum vulgare

561: !!! INVALID CITATION !!! . Change it.

591: Brassica oleracea

595: Hordeum vulgare

Author Response

(The authors gave the same response as above.)

Round 2

Reviewer 1 Report

The manuscript aims to describe the different technologies used to identify genes useful for barley breeding and some breeding techniques that use these technologies. The manuscript has been improved in some parts, but a further little effort can be made. More suggestions are added directly in the attached text, hoping that they will be useful to further improve the manuscript

Author Response

Dear reviewer,

Thank you for your valuable comments and suggestions which are all considered in the revised version of the manuscript. As per your suggestions, the improvements are done with track changes, and the modifications of comments are explained below in red-colored text;

The unnecessary words and sentences are removed, mistyped terms are corrected (line number given below with each section)

  • Abstract
    • Minor changes of sentence structure Line # 30-32
  • Introduction
    • Minor changes of sentence structure Line # 57, 61 66-68,70, 75, 77,78
  • 5. The indentation and numbering of 2.5 is corrected
    • Line # 110s
  • 1. QTL mapping
    • Minor changes of sentence structure Line # 60,61,63,69,77-79, 183-188
  • 2. GWAS
    • Thank you for your suggestions, the paragraphs are revised with minor changes of sentences structure Line 209-211, 226, 235-238
  • 3. Integration bridge
    • Thank you so much for your suggestion, we have written the limitations in each section (QTL; line 183-188 and GWAS; line 209-211) and the section of integration bridge to our come these limits is separated after both techniques; line 251.
  • 4. Genome-wide analysis
    • Minor changes of sentence structure Line 281-296
    • MAPK are in found in three different kinases, full word form of MAPK cascade is added (MAPK/KK/KKK; mitogen-activated protein kinase/kinase kinases/kinase kinase kinases)line 293
    • Sentence structure is changed to improve the relatedness of example Line # 298-304
  • Genome-based molecular breeding
    • Minor changes of sentence structure Line 310-311
    • As compared to rice and wheat, MAS is less applied in barley breeding for agronomic traits and stress resistance. So, suggestion to adapt the MAS based breeding in barley is given with recent successful example of MAS application for malting quality of barley (line 329-335)
    • Minor changes of sentence structure in line 361-363
  • Genome editing, characterization…….
    • Thank you for your suggestion, the paragraph is revised with regard of supporting traditional breeding by modern breeding technologies line 369-374
    • The full form of CRISPR/Cas9 is removed as it was already written in introduction line 66.
    • The suggested paragraph is moved from line 407 to line 393
    • Unnecessary words are removed from line 402-404
  • Conclusion
    • Sentence structure is improved from line 415, 422, and 429
  • Figure 3
    • The typos mistakes are removed
  • Resources
  • Modern
  • Whole-genome
  • Genome-editing
  • References
    • The references are added by using the endnote citation tool and the endnote format of MDPI style was downloaded from the link given in author’s instruction.